# Cerebral-Organoid-Derived Exosomes Alleviate Oxidative Stress and Promote LMX1A-Dependent Dopaminergic Differentiation

**DOI:** 10.3390/ijms241311048

**Published:** 2023-07-04

**Authors:** Xingrui Ji, Shaocong Zhou, Nana Wang, Jingwen Wang, Yue Wu, Yuhan Duan, Penghao Ni, Jingzhong Zhang, Shuang Yu

**Affiliations:** 1School of Biomedical Engineering (Suzhou), Division of Life Sciences and Medicine, University of Science and Technology of China, Hefei 230026, China; jxr2019@mail.ustc.edu.cn (X.J.); zhousc97@mail.ustc.edu.cn (S.Z.); wangnana93@mail.ustc.edu.cn (N.W.); dyh0624@mail.ustc.edu.cn (Y.D.); 2Suzhou Institute of Biomedical Engineering and Technology, Chinese Academy of Sciences, Suzhou 215163, China; wangjw@sibet.ac.cn (J.W.); wuy@sibet.ac.cn (Y.W.); niph@sibet.ac.cn (P.N.); 3School of Medical Imaging, Xuzhou Medical University, Xuzhou 221004, China

**Keywords:** cerebral organoids, exosomes, oxidative stress, dopaminergic differentiation, mesenchymal stem cells

## Abstract

The remarkable advancements related to cerebral organoids have provided unprecedented opportunities to model human brain development and diseases. However, despite their potential significance in neurodegenerative diseases such as Parkinson’s disease (PD), the role of exosomes from cerebral organoids (OExo) has been largely unknown. In this study, we compared the effects of OExo to those of mesenchymal stem cell (MSC)-derived exosomes (CExo) and found that OExo shared similar neuroprotective effects to CExo. Our findings showed that OExo mitigated H_2_O_2_-induced oxidative stress and apoptosis in rat midbrain astrocytes by reducing excess ROS production, antioxidant depletion, lipid peroxidation, mitochondrial dysfunction, and the expression of pro-apoptotic genes. Notably, OExo demonstrated superiority over CExo in promoting the differentiation of human-induced pluripotent stem cells (iPSCs) into dopaminergic (DA) neurons. This was attributed to the higher abundance of neurotrophic factors, including neurotrophin-4 (NT-4) and glial-cell-derived neurotrophic factor (GDNF), in OExo, which facilitated the iPSCs’ differentiation into DA neurons in an LIM homeobox transcription factor 1 alpha (LMX1A)-dependent manner. Our study provides novel insight into the biological properties of cerebral organoids and highlights the potential of OExo in the treatment of neurodegenerative diseases such as PD.

## 1. Introduction

Exosomes, a type of extracellular vesicle (EV) secreted by all cells, contain the cellular constituents of their host cells, including lipids, nuclei acids, proteins, metabolites, etc. Exosomes have been recognized as important mediators of intercellular communication, as they carry substances of the host cells that can be transferred to a recipient cell via the fusion of the exosome with the target cell membrane [1,2]. Exosomes have the unique capacity to cross the blood–brain barrier, thus making them promising candidates for the treatment of central nervous system (CNS) diseases [3,4]. Over the past few decades, numerous studies have evaluated the therapeutic potential of exosomes in the treatment of CNS diseases. Among these studies, exosomes derived from mesenchymal stem cells (MSCs) have been extensively investigated, as the therapeutic effects of MSCs are mainly mediated by the anti-inflammatory and trophic constituents of MSC-derived secretome. Studies have shown that MSC-derived exosomes (CExo) have neuroprotective and neurodegenerative effects, reduce neural inflammation, and promote the recovery of function after injury [5].

Exosomes are typically characterized by their size (40–160 nm) as well as the expression of a specific set of membrane and surface proteins, including CD9, CD63, CD81, etc. [2]. The composition of exosome cargoes and the information that they transport may vary depending on the cell of origin and its physiological state. Such cell-specific cargoes of proteins, lipids, and nuclei acids can be selectively taken up by neighboring or distant cells far from the donor cells, modulating the biological function of recipient cells. For example, the distinct exosomal protein patterns of integrin [6,7] and tetraspanin [8] contribute to the specificity of target cells, and different patterns of RNA cargoes elicit transient or persistent phenotypic changes in specific recipient cells [9]. Indeed, research has shown that neural stem cell (NSC)-derived EVs were more effective in improving cellular, tissue, and functional outcomes in a murine stroke model than MSC-derived EVs, even when both cell types were differentiated from the same iPSC cell line [10]. Moreover, neural-derived EVs are involved in the extensive cross-talk between neurons and glia in the CNS [11], transferring bioactive RNAs, proteins, and lipids between cells during synaptic activity to regulate neural and synaptic function [12]. Therefore, it is reasonable to suggest that exosomes from neural cells may hold great potential in the treatment of neurological disorders.

In recent years, cerebral organoids (COs) have been receiving increasing attention, largely due to their unique capacity to mimic the 3D architecture of complex, multi-layered brain tissue consisting of multiple neural cell types [13,14]. Publications have reported on the utility of human COs in studying molecular mechanisms and drug screening for CNS diseases [15,16]. However, the potential application of CO-derived exosomes (OExo) in regenerative medicine remains unclear. The different neural cell types present in COs, including neurons, astrocytes, and microglia, secrete exosomes that exert effects through various mechanisms. Astrocyte-derived exosomes, for example, exhibit neuroprotective effects through miR-190b and miR-200a-3p [17,18], growth factors such as fibroblast growth factor-2, and survival factors such as heat shock protein 70, etc. [19,20]. M2 microglia-derived exosomes attenuate ischemic brain injury and promote neuronal survival via miR-124 [21]. NSC-derived exosomes are enriched with specific miRNAs, exerting anti-inflammatory, neurogenic, and neurotrophic effects [22,23]. Investigating the impact of exosomes released from COs, which comprise various neural cell types, on neural cell function is, therefore, an area that warrants further exploration.

While specific mechanisms of action are still being investigated, there is no doubt about the potential therapeutic mechanisms of EVs for neurodegenerative diseases, including the regulation of oxidative stress and neural plasticity. In the present study, we sought to evaluate the neuroprotective effects and differentiation capacity of OExo, in addition to comparing their function to CExo. Understanding the biological activity and underlying mechanisms of OExo would provide valuable insights for the development of novel strategies for treating neurodegenerative diseases such as Parkinson’s disease (PD).

## 2. Results

### 2.1. Cerebral Organoids (COs) Comprise Neuronal and Glial Phenotypes

The dissociated iPSCs reaggregated to form embryoid bodies (EBs), which gradually showed evidence of ectodermal differentiation in the induction medium after 7 days (Figure 1(A1)). The EBs were distinguishable by the brightened surface tissue and the dark center with dense non-ectodermal tissue (Figure 1(A2)). After being embedded in Matrigel, the organoids continued to proliferate, exhibiting neuroepithelial bud outgrowth on the 20th day (Figure 1(A3)). By Days 40–60, typical mature COs could be observed, featuring optically clear neuroectoderm as well as large buds of translucent ectoderm that were not radially organized (Figure 1(A4)).

Starting from Day 60 onwards, the COs displayed a dense ventricular zone (VZ) populated with neural progenitors expressing paired box 6 (PAX6), which was surrounded by chicken ovalbumin upstream promoter transcription factor (COUP-TF)-interacting protein 2 (CTIP2)+ cortical neurons (Figure 1B). Similar to human cortical organization, the eomesodermin (TBR2)+ intermediate progenitors were located adjacent to the early cortical plate, which was marked with CTIP2+ cells (Figure 1C). At this stage, several brain regions were visible, including the forebrain marked with forkhead box G1 (FOXG1) staining (Figure 1D) and the hippocampus marked with frizzled class receptor 9 (FZD9) staining (Figure 1E). Various glia cells were also observable in the COs, as indicated via the staining of the induction of ionized calcium-binding adapter molecule 1 (IBA-1) for microglia (Figure 1F) and glial fibrillary acidic protein (GFAP) for astrocytes (Figure 1G). Collectively, these results indicated the successful construction of COs containing neuronal and glial identities.

### 2.2. Characterization of OExo and CExo

MSCs have been characterized via flow cytometry, showing high enrichment in the mesenchymal markers cluster of differentiation 90 (CD90) and CD105 and the absence of the hematopoietic markers CD34 and CD45 (Appendix A).

In the study, exosomes were successfully isolated from MSCs (P3–P6) (CExo) and COs (60–140 days) (OExo) using sequential centrifugations (Figure 2). Bilayer membrane vesicles were observed in both CExo and OExo using TEM (Figure 3A,B). The presence of exosomal markers, including CD9, CD63, and TSG101, in both CExo and OExo was confirmed using Western blot (WB) (Figure 3C). Nanoparticle tracking analysis revealed that the CExo had an average diameter of 136.6 nm and a particle concentration of 2.4 × 10^9^ particles/mL (Figure 3D); similarly, the OExo had an average diameter of 131.2 nm and a particle concentration of 7.9 × 10^9^ particles/mL (Figure 3E). A total of 160 mL supernatants were collected from 6 × 10^7^ MSCs cultured in conventional 2D flasks. The COs had an average diameter of 2–3 mm containing ~2 × 10^6^ cells after 60 days. Thirty COs (total ~6 × 10^7^ cells) were cultured and 200 mL of supernatants was collected between Days 60 and 140. Particle quantification analysis revealed that the OExo yields were approximately three times higher than the CExo ones with similar cell densities.

To evaluate the uptake of CExo and OExo by astrocytes and iPSCs in vitro, the CExo or OExo were labeled with PKH26, a fluorescent cell linker compound. The labeled CExo or OExo were efficiently taken up by rat astrocytes (Figure 3F) or iPSCs (Figure 3G) within 48 h, as evidenced by the scattered red fluorescence around the cell nuclei. All of these results demonstrated that OExo and CExo were successfully purified, and thus were suitable for further studies.

### 2.3. OExo and CExo Rescued H_2_O_2_-Induced Oxidative Stress in Astrocytes

EVs derived from MSCs are known to inhibit reactive oxygen species (ROS) production and inflammation [24]. Here, we aim to evaluate whether OExo have similar neuroprotective effects on astrocytes, which play a crucial role in the regulation of ROS in the CNS [25]. The purity of astrocytic culture was identified by the expression of GFAP in 90% of the cells (Appendix A). H_2_O_2_ induced a dose-dependent reduction in cell viability in the astrocytic culture (Figure 4A), and the viability was reduced to 75% when H_2_O_2_ was applied at 20 µM, which was the dosage selected for all of the following experiments. The CExo and OExo significantly rescued H_2_O_2_-impaired viability in astrocytes (Figure 4C), while having no impact on cell viability when used alone (Figure 4B). In line with these observations, 20 µM of H_2_O_2_ treatment significantly increased the ROS levels in the astrocytes, as evidenced by the increased fluorescent signal of Dihydroethidium (DHE), an ROS indicator (Figure 4D,E). Importantly, the H_2_O_2_-increased ROS levels could be significantly reduced via the treatment of CExo or OExo (Figure 4D,E). Oxidative stress occurs when the production of free radicals exceeds the antioxidant capacity of cells. Consequently, H_2_O_2_ was shown to significantly reduce the activity of superoxide dismutase (SOD), one of the main antioxidant enzymes, and the CExo and OExo showed comparable rescuing effects on the H_2_O_2_-induced SOD depletion (Figure 4F).

ROS production resulted in lipid peroxidation, mitochondrial perturbation, and apoptotic cell death [26]. Malondialdehyde (MDA), a product of lipid peroxidation, was significantly increased via H_2_O_2_ treatment, and such an increase was significantly alleviated by either CExo or OExo exposure (Figure 5A). In line with these observations, H_2_O_2_ treatment resulted in a loss of mitochondrial membrane potential (ΔΨm), which was evidenced by the reduced fluorescent ratio (red/green) of JC-1, a ΔΨm-sensitive fluorescent probe. Moreover, both the CExo and OExo significantly rescued H_2_O_2_-induced ΔΨm loss (Figure 5B). Accordingly, the increased ratio of BCL-2-associated X protein (BAX) to B-cell lymphoma-2 (BCL-2), which was suggested to induce a collapse of mitochondrial membrane potential, was observed at transcriptional (Figure 5C) and translational (Figure 5D,E) levels via H_2_O_2_ exposure, and such a high ratio was significantly reduced by either CExo or OExo treatment (Figure 5C–E). In addition, CExo and OExo exposure significantly alleviated the H_2_O_2_-activated mitochondrial apoptotic pathway, as evidenced by the reduced expression levels of apoptotic protease activating factor-1 (*apaf-1*) and *p53*, as well as the apoptotic executor enzyme *caspase3* (Figure 5F). Taken together, these results suggest that OExo share similar neuroprotective effects to CExo in reducing cellular oxidative stress.

### 2.4. OExo Promote Dopaminergic Differentiation of iPSCs

The iPSCs were differentiated into neuron-specific class III β-tubulin (TUJ1)+ neurons with faint tyrosine hydroxylase (TH) expression when cultured in the basal neuronal induction medium (NM) for 14 (Figure 6A) or 28 (Figure 6C) days, as revealed via double immunostaining against TUJ1 and TH, the markers of post-mitotic and DA neurons, respectively. When compared to the control (Ctrl), the addition of CExo to NM significantly enhanced the expression levels of TUJ1 by Day 14 (Figure 6A,B,E). However, no increase in TH expression was observed, even when the differentiation period was extended to 28 days (Figure 6C–E). These findings suggest that CExo might promote non-DA neural lineage differentiation. In contrast, iPSCs differentiated with the addition of OExo generated DA neurons positive for both TH and TUJ1, and such DA neurons exhibited mature neuron morphology with complex neurites in the culture differentiated for 28 days (Figure 6A,C). An analysis of fluorescent signal and WB showed that iPSCs differentiated with the addition of OExo for 14 or 28 days produced significant increased expression levels of TH and TUJ1, when compared to the Ctrl or the one treated with CExo (Figure 6B,D,E). These data indicate that OExo are superior to CExo regarding the induced differentiation of iPSCs into DA neurons.

### 2.5. OExo Promote iPSC Differentiation into Dopaminergic Neurons via LMX1A Pathway

DA differentiation was regulated by a series of orchestrated transcriptional factors (TFs) [27]. Using RT-PCR to screen the expression levels of a number of key TFs during DA differentiation, we found that the LIM homeobox transcription factor 1 alpha (*lmx1a*) was specifically upregulated in cells treated with OExo at both the transcriptional (Figure 7A) and translation (Figure 7B) level, as compared to the Ctrl or the one treated with CExo. It was noteworthy that *wnt1*, which forms an autoregulatory loop with *lmx1a* [28], was also significantly increased in the OExo-treated culture in comparison with those treated with the Ctrl or CExo (Figure 7C). Other TFs, such as midbrain transcription factor engrailed homeobox 1 (*en1*), serine peptidase (*corin*), mammalian achaete-scute homolog-1 (*mash1*), polypyrimidine-tract-binding protein 1 (*ptbp1*), paired-like homeodomain 3 (*pitx3*), and nuclear-receptor-related protein 1 (*nurr1*), exhibited similar or insignificant expression patterns to the iPSCs treated with CExo and OExo (Figure 7A).

To investigate the role of LMX1A in the OExo-promoted DA differentiation of iPSCs, GTX31507-PEP (GTX), an LMX1A-blocking peptide, was applied in an OExo-treated differentiation system. The blockage of the LMX1A pathway was confirmed via immunostaining (Figure 7D) and WB (Figure 7E), showing complete abolishment of LMX1A expression in the neural culture differentiated from the iPSCs. Concomitantly, TH expression was significantly reduced in these LMX1A-blocked neural cultures (Figure 7D,E), suggesting that OExo-promoted DA differentiation from iPSCs is crucially regulated by LMX1A expression.

DA differentiation is regulated by various neurotrophic factors (NFs) [29]. Using RT-PCR and ELISA, we revealed a striking difference in the NF expression patterns released by MSCs vs. COs (Figure 7F,G). The mRNA levels of NT-4 and glial-cell-line-derived neurotrophic factor (GDNF) were expressed approximately 50 times and three times higher in COs, respectively, compared to the corresponding levels in MSCs. In contrast, the brain-derived neurotrophic factor (BDNF) level of COs was about half of that of MSCs (Figure 7F). ELISA further confirmed that the OExo had an approximately 20 times higher expression of NT-4 than that of the CExo (Figure 7G). Such highly enriched NFs in the COs probably underlie the mechanisms of OExo-promoted LMX1A upregulation and DA differentiation.

## 3. Discussion

In this study, we examined the neuroprotective properties of OExo and their potential to induce the DA phenotype. Both OExo and CExo displayed similar antioxidant and neuroprotective effects in the astrocytic culture, alleviating excess ROS production, lipid peroxidation, mitochondrial dysfunction, and the expression of apoptotic genes. Moreover, OExo were found to be enriched with more NFs, such as NT-4 and GDNF, in comparison to CExo, which facilitated iPSC differentiation into DA neurons through LMX1A upregulation. Our study first revealed the potential application of OExo in the treatment of neurodegenerative diseases such as PD.

Oxidative stress is a critical contributor to cellular injury, inflammation, and metabolic dysregulation, which are involved in many pathologies. Our study demonstrated that OExo alleviated H2O2-induced oxidative stress in astrocytes through various mechanisms, including the direct scavenging of ROS production and lipid peroxidation, the promotion of antioxidant defense, the stabilization of mitochondrial bioenergetics, and reduced sensitivity to apoptosis. These observations are in accordance with the reports that exosomes derived from various neural cells, including astrocytes [17], microglia [21], and NSCs [22], exhibit antioxidant and neuroprotective effects. While the underlying mechanisms are not fully understood, various exosomal bioactive molecules, including antioxidative enzymes, trophic factors, miRNAs, and circRNAs, have been reported to contribute to the therapeutic effects of neural-cell-derived exosomes [30]. It is conceivable that similar and multiple regulatory mechanisms underlie the antioxidant and cytoprotective effects of OExo. Furthermore, the antioxidant properties of MSCs have been previously associated with their cytoprotective and anti-inflammatory effects across species and spectra of disease [31]. ROS are key signaling molecules in the progression of inflammatory disorders [32]. Accumulating evidence suggests a critical role of inflammatory mechanisms in a number of neurological diseases, such as Alzheimer’s disease, Parkinson’s disease, Huntington’s disease, etc. [33]. Neurodegenerative processes are fueled by neuroinflammation, which is often accompanied by systematic immune dysregulation. Given that OExo and CExo exert similar antioxidant properties, our study implies the potential applicability of OExo in the treatment of neurological disorders. Although up until now most studies on COs have focused on disease modeling and drug screening, recent studies have reported that exosomes from iPSC-derived retinal organoids [34], gut organoids [35] and skin organoids [36] exhibit anti-inflammatory potential, indicating the novel application of COs and their exosomes in terms of inflammation modulation.

PD is characterized by a loss of DA neurons in the substantia nigra, and being supplemented with DA neurons differentiated from iPSCs is an important strategy for PD therapy. To achieve high efficiency of DA phenotype induction, both intrinsic transcriptional regulation and external microenvironment modulation have to be considered [37]. The observation that OExo are superior to CExo regarding the induced differentiation of iPSCs into DA neurons reinforces the potential application of OExo in the treatment of PD. By utilizing OExo, a favorable microenvironment for the induction of DA neurons might be provided.

The development of the DA phenotype is controlled by a series of TFs. Through screening for pivotal TFs, we found that *lmx1a* was significantly upregulated by OExo and a blockade of LMX1A expression was sufficient to deplete OExo-induced DA differentiation. *Wnt1*, which forms an autoregulatory loop with *lmx1a* during DA differentiation, was also upregulated by OExo application in the differentiation system. These observations suggest that OExo-induced DA differentiation is dependent on the proper activation of *lmx1a*. It is not surprising that *lmx1a* plays such an essential role in DA phenotype induction. It is a homeodomain transcription factor, playing a key role in the specification of DA neurons in vivo [38]. When exogenously expressed in embryonic stem cells or induced NSCs, *lmx1a* markedly enhanced the efficiency of DA neuron production [39,40].

As discussed previously, exosomes contain complex components. In terms of the neurotrophic factors that are closely related with DA development, OExo were found to contain high levels of NT-4 and GDNF. NT-4 shares receptors with BDNF, binding to the TrkB receptor and triggering the phosphatidylinositol 3-kinase-Protein Kinase B (PI3K-Akt) and phospholipase C-gamma1 pathways [41]. Compared to other NFs such as BDNF and neurotrophin-3 (NT-3), NT-4 has been less extensively studied; however, it has been reported to be more efficient in inducing DA neurons in ventral mesencephalic culture systems [42]. Moreover, NT-4 protected the survival of DA neurons exposed to various stimuli in vitro and in vivo [43]. Although we failed to find direct evidence of NT-4 impacting LMX1A, such a highly enriched and distinct expression of NT-4 in OExo would undoubtedly be involved in LMX1A-dependent DA induction. Further research is needed to fully understand the molecular mechanisms underlying the effects of NT-4 on DA development. Moreover, the combination of NT-4 and GDNF has been reported to increase the survival and number of DA neurons and the dopamine content in ventral mesencephalic culture [44]. Therefore, it is reasonable to assume that the specific NF pattern in OExo offered a favorable microenvironment for iPSCs to adopt a DA identity.

In addition, it is worth noting that the COs were cultured in a 3D system, and the MSCs were cultured in a 2D system. This might explain the higher yield of OExo compared with CExo from a similar number of plating cells. However, the dimensional difference should not affect the cellular identity, as it would only impact the amount of released exosomes rather than the category of exosomes, having an impact on the amount rather than category of exosomes.

Overall, our findings demonstrate that OExo exhibited similar neuroprotective effects to CExo; however, they were superior in terms of DA induction properties. Our study has expanded the application of brain organoids beyond disease simulation and drug screening, and offered a novel strategy for PD therapy. Meanwhile, although we have shown that enriched NT-4 in OExo might be involved in LMX1A-dependent DA differentiation, further investigation using comparative proteomics and RNA-seq techniques is needed to fully understand the biomolecules and pathways that underlie OExo-promoted DA induction.

## 4. Materials and Methods

### 4.1. Generation of Cerebral Organoids (COs) from iPSCs Culture

Human iPSCs (kindly provided by Guangzhou Stem cell and Regenerative Laboratory, China) were cultured in Essential 8 (E8) medium (Stem Cell, Vancouver, BC, Canada) with pre-coated Matrigel (Corning, New York, NY, USA) as previously described [45]. COs were prepared as described by Lancaster et al. [46]. Briefly, iPSCs were in EB formation medium [46] for 7 days, and then transferred to the induction medium [46] for another 7 days. On the 15th day, organoids were started to expand by culturing in the expansion medium [46] in Matrigel, followed by culture in the maturation medium [46] on an orbital shaker (Wiggens, Straubenhardt, Germany; WOS-101SRC) from the 20th day. The medium was changed every 7 days afterwards. COs were assayed for immunostaining on the 60th, 80th, 100th, 120th, and 140th after iPSCs differentiation. The culture supernatant from mature COs, which exhibited stable neuronal phenotypes between 60 and 140 days [46], was collected for the preparation of OExo. 

### 4.2. Primary Culture of MSCs

All studies involved with human samples were performed in accordance with the Ethical Guiding Principles on Human Embryonic Stem Cell Research’ (of the Ministry of Science and Technology and the Ministry of Health, China, 2003) and Helsinki Declaration. Umbilical cords issues were generously donated with the signed informed consent and ethical approval from the first affiliated hospital of Soochow University. MSCs were isolated and cultured as previously described [47,48]. Briefly, the umbilical cord vessels and outer membrane were removed, and mesenchymal tissue in Wharton’s jelly was dissected and minced into 0.5 cm^3^ pieces. The small pieces were inoculated in 10 cm diameter culture dishes covered with DMEM/F12 medium (Gibco, Baltimore, MD, USA) containing 10% fetal bovine serum (FBS; Gibco, USA). MSCs were characterized by flow cytometry for surface markers specifically labeling mesenchymal (CD90 and CD105) and hematopoietic (CD34 and CD45) stem cells as previously described [49].

### 4.3. Exosomes Purification and Characterization

The culture medium from COs, collected between 60 and 140 days of culture, and the medium from MSCs during the 3rd to 6th passages (P), were acquired and stored at −80 °C until exosome isolation. The collected medium was subjected to sequential centrifugation steps at different speeds, starting at 300× *g* for 10 min to remove residual cells, then 2000× *g* for 10 min to remove cell debris, and 10,000× *g* for 30 min to eliminate apoptotic bodies using a Sorvall LYNX 6000 high-speed centrifuge (Thermo Fisher Scientific, Waltham, MA, USA). Finally, the supernatant was centrifuged at 120,000× *g* for 70 min using a Beckman Optima LX-80 ultracentrifuge (Beckman Coulter, Brea, CA, USA). The resulting pellets were resuspended in cold phosphate-buffered saline (PBS; Gibco, USA) and stored at −80 °C (Figure 2). Exosome preparations were verified by transmission electron microscopy (TEM). Exosome size and particle number were analysed using the LM10 or DS500 nanoparticle characterization system (Malvern Instruments, Malvern, UK; NanoSight) equipped with a blue laser (405 nm). The protein content of exosomes was assessed by the bicinchoninic acid (BCA) assay (Beyotime, Shanghai, China). For exosome-tracking purposes, purified exosomes were fluorescently labeled using PKH26 membrane dye (Sigma, St. Louis, MO, USA). Labelled exosomes were washed in 20 mL of PBS, collected by ultracentrifugation and resuspended in PBS.

### 4.4. Oxidative Stress to the Rat Midbrain Astrocytes

The primary astrocytic culture was prepared from 3-to-5-days-postnatal Sprague–Dawley rats (SPF Biotechnology Co., Beijing, China) as described previously [37]. Astrocytes (P3–P6) was pre-treated with H_2_O_2_ at various concentrations for 12 h. After washed out, cells were treated with OExo or CExo at a final concentration of 100 μg/mL for another 48 h (Figure 2). The astrocytes receiving PBS were set as control (CON).

### 4.5. Assay of SOD Activity and MDA Concentration

Superoxide dismutase (SOD) activity and malondialdehyde (MDA) concentration were determined using the total superoxide dismutase (T-SOD) assay kit (Nanjing Jiancheng Bioengineering Institute, Nanjing, China) and lipid peroxidation MDA assay kit (Beyotime, China), respectively. The absorbance at 550 nm or 532 nm were determined for SOD or MDA assay on a multimode microplate reader (Thermo Fisher Scientific, USA; Varioskan LUX).

### 4.6. ROS Measurement

Reactive oxygen species (ROS) were measured with the Dihydroethidium (DHE; Beyotime, China) [50]. DHE were added to the astrocytic culture at 5 μM and incubated for 45 min at 37 °C. The cell nuclei were visualized by staining with 4′,6-diamidino-2-phenylindole (DAPI; Beyotime, China) at 0.5 µg/mL for 15 min. Cells were observed under a confocal microscope (Nikon, Tokyo, Japan; A1R HD25). 

### 4.7. Determination of the Mitochondrial Membrane Potential (ΔΨm)

The dissociated cells were incubated with 5,5′,6,6′-Tetrachloro-1,1′,3,3′-tetraethyl-imidacarbocyanine iodide (JC-1; Beyotime, China) for 30 min. Subsequently, cells were washed twice with PBS and the fluorescence was measured at 530/590 nm with the excitation at 490/525 nm on the multimode microplate reader. The calculation of ΔΨm was performed by measuring the ratio of readings at 590 nm (red fluorescent J-aggregates) to those at 530 nm (green monomeric).

### 4.8. Dopaminergic Differentiation from the iPSCs

The neuronal induction medium (NM) [37] containing Neurobasal medium, 1% N2 supplement (Gibco, USA), 2% B27 supplement with vitamin A (Gibco, USA), 1% Gluta-MAX (Gibco, USA), 10 μM Y-27632 (Abmole, Houston, TX, USA) and 150 µM ascorbic acid (Peprotech, Cranbury, NJ, USA) was used as the basal medium for iPSCs differentiation into DA neurons. iPSCs were differentiated in either NM (Ctrl), NM supplemented with 100 μg/mL CExo (CExo) or 100 μg/mL OExo (OExo) for 14 or 28 days (Figure 2). For LMX1A blocking experiments, 0.5 μg/mL LMX1A blocking peptide (GeneTex, Irvine, CA, USA; GTX31507-PEP) (GTX) were added 1 h before the OExo treatment. The medium was changed and the corresponding supplements were replenished every 3 days during the differentiation process.

### 4.9. Immunostaining

The organoids were fixed in 4% paraformaldehyde (PFA; Solarbio, Beijing, China) at 4 °C overnight, dehydrated with 30% sucrose (Solarbio, China) solution for 2 days, followed by embedding in 30% sucrose solution containing 7.5% gelatin (Solarbio, China) on dry ice in the base molds. 30 μm of cryosections were prepared, blocked with 5% BSA in 0.2% Triton-100 (Solarbio, China) for 2 h, and then incubated with the following primary antibodies overnight at 4 °C: anti-GFAP (Abcam, Cambridge, UK; ab7260; 1:1000), anti-PAX6 (Abcam, UK; ab195045; 1:500), anti-IBA-1 (Wako, Osaka, Japan; 019-19741; 1:1000), anti-FOXG1 (Abcam, UK; ab196868; 1:500), anti-CTIP2 (Abcam, UK; ab18465; 1:500), anti-FZD9 (SAB, Baltimore, MD, USA; 31503; 1:500), and anti-TBR2 (Abcam, UK; ab216870; 1:1000). The cells were incubated with the following primary antibodies: anti-LMX1A (Sigma, USA; ab10533; 1:2000), anti-TUJ1 (Abcam, UK; ab78078; 1:1000), and anti-TH (Abcam, UK; ab113; 1:1000). Immunoreactivity was visualized using appropriate Alexa Fluor-conjugated secondary antibodies and observed using the confocal microscope (Nikon, Tokyo, Japan, A1R HD25). For both organoids and cells, DAPI were counterstained at 0.5 µg/mL for 15 min. The fluorescence intensity was analyzed by ImageJ (National Institutes of Health, NIH, Bethesda, MD, USA), and performed on 3–5 individual microscopic fields, randomly chosen across 2 diameters of each coverslip (×400 magnification).

### 4.10. Western Blot

Cells or exosomes were lysed in the cell lysis buffer (Beyotime, China), separated and incubated with the following antibodies: anti-TUJ1 (1:1000), anti-TH (1:1000), anti-CD63 (Abcam, UK; ab134045; 1:200), anti-CD9 (CST, Danvers, MA, USA; D8O1A; 1:200), anti-TSG101 (Santa Cruz, Heidelberg, Germany; sc-7964; 1:200), anti-LMX1A (1:1000), anti-BAX (CST, USA; 14796S; 1:1000), anti-BCL-2 (CST, USA; 3498S; 1:300), and anti-β-actin (Santa Cruz, Heidelberg, Germany; sc-8432; 1:1000). Specific protein bands were visualized by enhanced chemiluminescence (Thermo Fisher Scientific, USA), and quantified with the Image Pro Plus software version 6.0 (Media Cybernetics, Bethesda, MD, USA).

### 4.11. Quantitative PCR

mRNA was extracted with Total RNA Kit I (Omega Bio-Tek, Norcross, GA, USA) and reverse-transcribed into cDNA using Prime Script RT reagent kit (TAKARA Bio, Kusatsu, Japan). Quantitative PCR was performed on Applied Biosystems QuantStudio Real-Time PCR System (Thermo Fisher Scientific, USA) by using TB Green Premix Ex Taq (TAKARA Bio, Japan) and 2^−ΔΔCT^ method. The primers were described as Appendix A; *gpadh* and *β-actin* was used as internal controls.

### 4.12. Statistical Analysis

Numerical data are presented as mean ± standard deviation (SD). The data were subjected to unpaired t-tests or analysis of variance and an appropriate post-hoc analysis using GraphPad Prism version 8.0 (GraphPad Software, San Diego, CA, USA). The level of significance was set at *p* < 0.05.

## Figures and Tables

**Figure 1 ijms-24-11048-f001:**
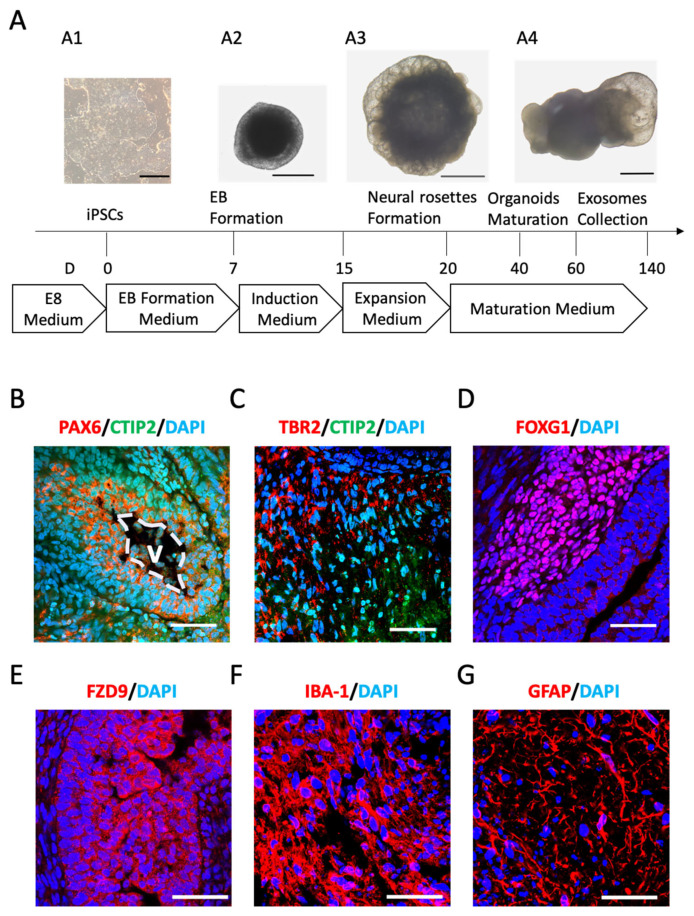
Generation and characterization of cerebral organoids (COs). (**A**) Schematic illustration of COs, depicting the crucial stages of the differentiation process. On Day (D) 0, a colony of iPSCs with defined boundaries could be observed (**A1**). Embryoid bodies (EBs) were formed in EB formation medium on D7, which exhibited brightened surface tissue and a dark center (**A2**). Neural rosettes developed on D20, displaying a radially tubular structure in the induction and expansion medium (**A3**). Mature COs with neuroepithelial bud outgrowth appeared on D40 after being transferred to maturation medium (**A4**). Exosomes were collected from the culture medium of COs during D60-140. (**B**–**G**) Representative staining for brain regions, neuronal, and glial cell identities on D100. DAPI was used for nuclear counterstaining. (**B**) Staining for neural progenitors (PAX6) and the early-born cerebral deep-layer neurons (CTIP2). The cortical regions developed a typical ventricular zone (V) surrounded by PAX6+ neural progenitors. (**C**) Staining for the intermediate progenitors (TBR2) and the early-born cerebral deep-layer neurons (CTIP2). Note that TBR2+ progenitors were expressed adjacent to CTIP2+ neurons. (**D**) A forebrain region of an organoid staining positive for the marker FOXG1. (**E**) Hippocampal regions that stained positive for the marker FZD9. (**F**) Staining for microglia marker IBA-1. (**G**) Staining for astrocytic marker GFAP. Scale bars: 500 μm in (**A1**), 1000 μm in (**A2**–**A4**), and 50 μm in (**B**–**G**).

**Figure 2 ijms-24-11048-f002:**
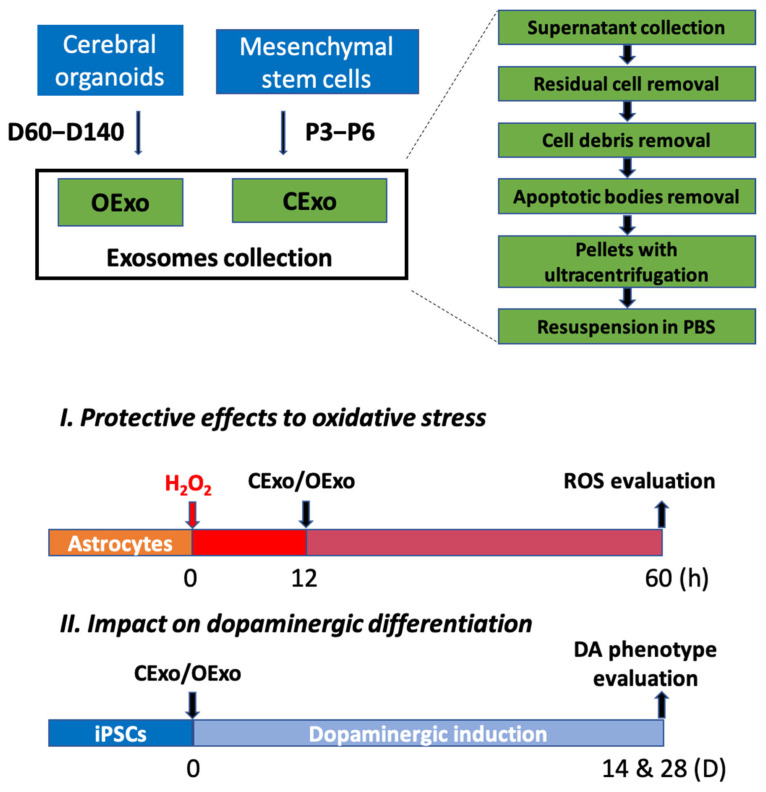
The experimental design and workflow of the study. Cerebral-organoid-derived exosomes (OExo) were collected from Day (D) 60 to D140 after iPSC differentiation, while the mesenchymal-stem-cell-derived exosomes (CExo) were collected from Passage (P)3 to P6. The culture medium was collected, followed by sequential centrifugation (200× *g* for 10 min to remove residual cell → 2000× *g* for 10 min to remove cell debris → 10,000× *g* for 30 min to remove apoptotic bodies → 120,000× *g* for 70 min to pellet the exosomes) to obtain the corresponding exosomes. To evaluate the protective effects against H_2_O_2_-induced oxidative stress, OExo or CExo were added to purified astrocytic culture 12 h after H_2_O_2_ treatment, and ROS production, lipid peroxidation, mitochondrial activity and, apoptosis were assessed 48 h after the addition of exosomes. To test the exosomes’ impact on dopaminergic (DA) differentiation, the iPSCs were differentiated under a basic DA induction system with the addition of OExo or CExo for either 14 d or 28 d, and the DA phenotype as well as the underlying mechanism were analyzed.

**Figure 3 ijms-24-11048-f003:**
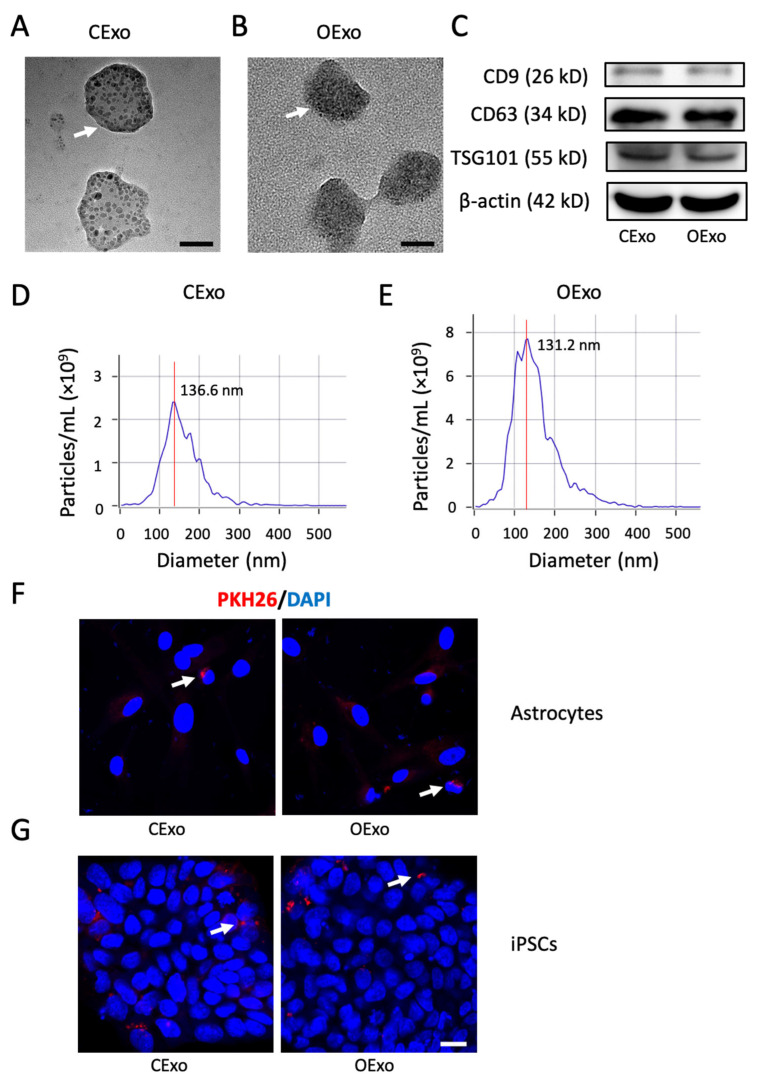
Characterization of exosomes from cerebral organoids (OExo) or MSCs (CExo) via ultracentrifugation. (**A**,**B**) Representative TEM images of the isolated CExo (**A**) and OExo (**B**). The white arrows indicate the bilayer membrane of exosomes. (**C**) Western blot analysis confirming the expression of CD9, CD63, and TSG101 in both CExo and OExo, using β-actin as an internal control. The images are the representatives of three independent experiments. (**D**,**E**) The size distribution profiles of the isolated CExo (**D**) and OExo (**E**) measured via NTA. (**F**,**G**) Representative images of PKH26 fluorescently labeled CExo and OExo uptake by astrocytes (**F**) and iPSCs (**G**) after 48 h of incubation. The white arrows indicate the exosome uptake around the DAPI-stained nuclei. Scale bars: 50 nm in (**A**,**B**), 20 μm in (**F**,**G**).

**Figure 4 ijms-24-11048-f004:**
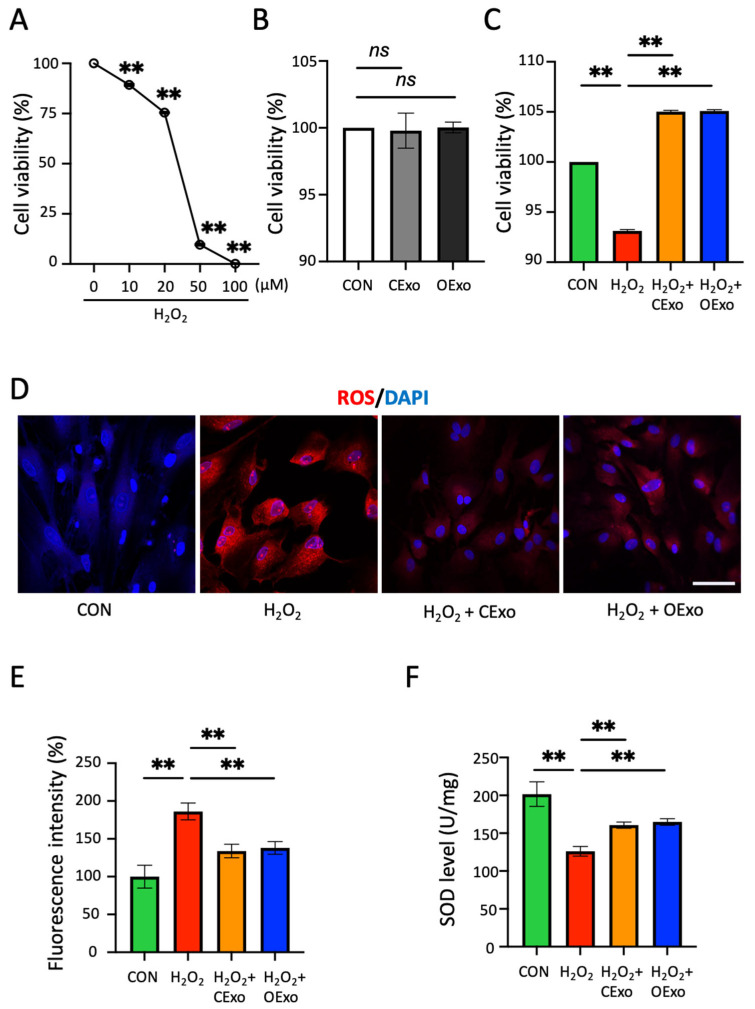
OExo and CExo alleviated H_2_O_2_-induced ROS and antioxidant depletion in astrocytes. (**A**) Astrocytes were incubated with H_2_O_2_ concentrations ranging from 0 to 100 μM for 12 h, and the cell viability was analyzed using the CCK-8 assay. H_2_O_2_ produced a significant reduction in cell viability with 10 μM concentration. (**B**) Astrocytes were incubated with 100 μg/mL CExo or OExo for 48 h, and the cell viability analysis was evaluated using the CCK-8 assay. (**C**) Astrocytes were incubated with 20 μM H_2_O_2_ for 12 h and then treated with 100 μg/mL CExo or OExo for 48 h. Cell viability assay showed that CExo and OExo significantly rescued H_2_O_2_-induced cell loss. (**D**) Treatment of astrocytic cultures with 20 μM H_2_O_2_ stimulated ROS production, as indicated via the intercalation of ethidium into DNA (red fluorescence). CExo and OExo alleviated ROS production post-treatment. (**E**) ROS fluorescent intensity analysis showed that the CExo and OExo significantly rescued H_2_O_2_-stimulated ROS production, as measured via dihydroethidium (DHE) staining. (**F**) Treatment of culture with H_2_O_2_ led to a significant reduction in the key antioxidant enzyme superoxide dismutases (SOD), and CExo and OExo significantly alleviated such a reduction post-treatment. Enzyme activities were normalized to protein concentrations of the cell extracts. Mean ± SD. ns indicates non-significant, ** indicates *p* < 0.01 vs. control (CON), *n* = 3 biological replicates per group for (**A**–**C**) and (**F**); *n* = 9 fields (3 fields per biological replicate × 3 biological replicates per group) for (**E**). Scale bar: 50 μm.

**Figure 5 ijms-24-11048-f005:**
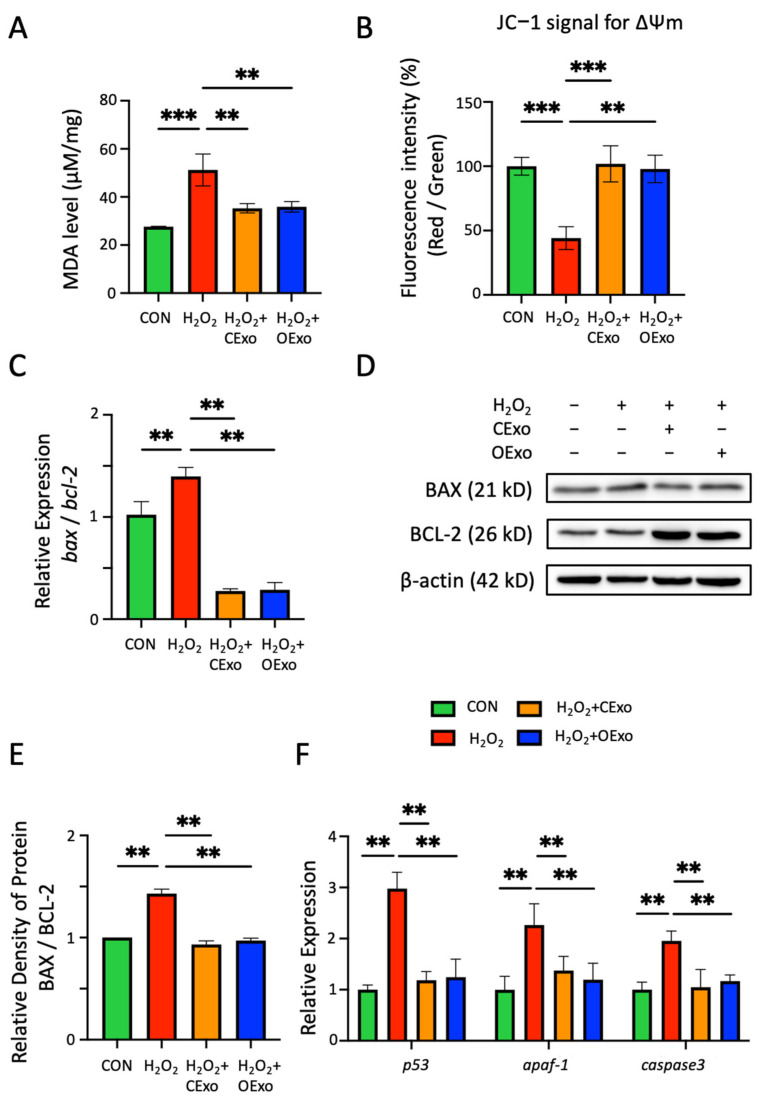
The protective effects of CExo and OExo against lipid peroxidation, mitochondrial dysfunction and apoptosis in astrocytes. (**A**) Malondialdehyde (MDA), one critical final product of polyunsaturated fatty acid peroxidation, was significantly upregulated via H_2_O_2_ treatment. CExo or OExo application significantly alleviated an H_2_O_2_-induced MDA increase. (**B**) CExo or OExo treatment significantly rescued H_2_O_2_-induced mitochondrial membrane potential (ΔΨm) loss. ΔΨm alteration in astrocytes was determined by fluorescent signals at 590/530 nm (red/green ratio) after staining with JC-1. (**C**) RT-PCR revealed that H_2_O_2_ treatment significantly increased the ratio of pro-apoptotic *bax* to anti-apoptotic *bcl-2* mRNA levels. CExo or OExo application significantly reduced H_2_O_2_-upregulated *bax*/*bcl-2* ratios. (**D**) Representative immunoblotting images showing the expression levels of BAX (21 kD) and BCL-2 (26 kD) in astrocytes which were treated with H_2_O_2_ in the presence or absence of CExo and OExo. β-actin (42 kD) was used as an internal control. (**E**) Semi-quantitative analysis revealed that CExo or OExo treatment significantly reduced H_2_O_2_-upregulated BAX/BCL-2 ratios. (**F**) RT-PCR data showing the mRNA levels of *p53*, *apaf-1* and *caspase3* in astrocytes treated with H_2_O_2_ in the absence or presence of CExo and OExo. Mean ± SD. ** indicates *p* < 0.01, *** indicates *p* < 0.001. *n* = 3 independent biological replicates for each group.

**Figure 6 ijms-24-11048-f006:**
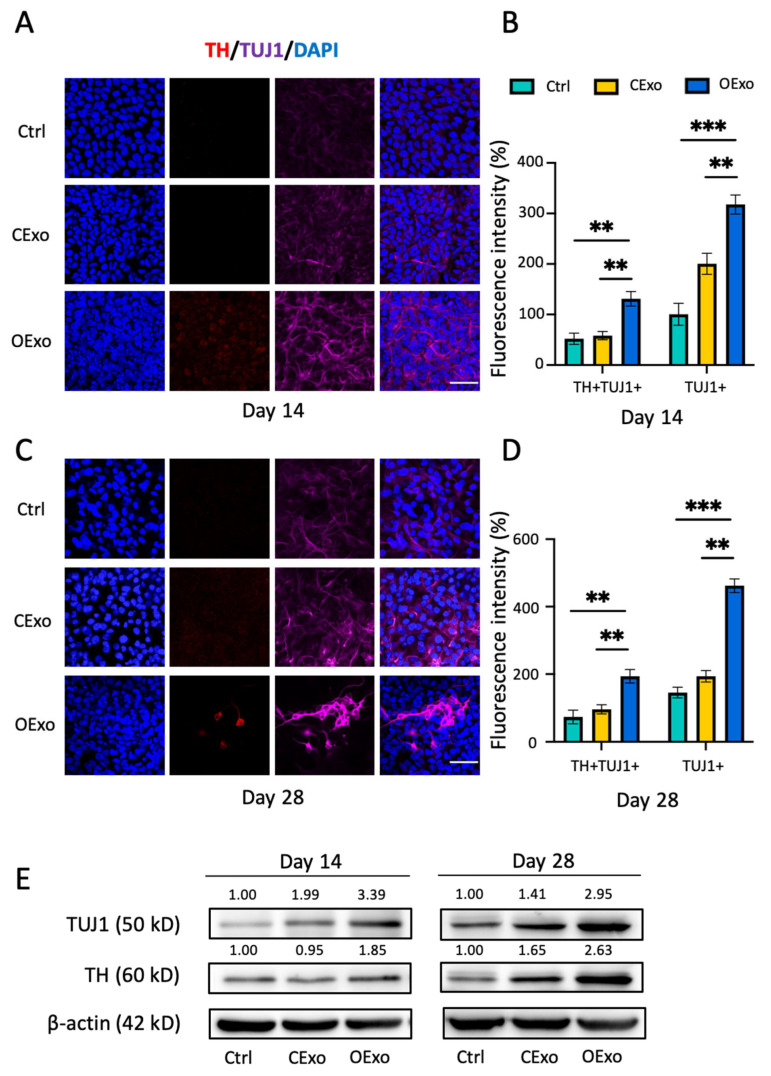
OExo but not CExo promoted iPSC differentiation into DA neurons. (**A**,**C**) Representative immunostaining images showing that iPSCs were differentiated in either neuronal induction medium (NM; Ctrl), NM with CExo (CExo), or NM with OExo (OExo) for 14 days (**A**) or 28 days (**C**), and were then subjected to double staining for TH and TUJ1. DAPI was used for counterstaining. (**B**,**D**) Semiquantitative analysis of TH and TUJ1 fluorescent intensities showed that iPSCs differentiated in the presence of OExo produced significantly more TH+TUJ1+ DA and TUJ1+ neurons, as compared to iPSCs differentiated in NM or CExo for 14 days (**B**) and 28 days (**D**). (**E**) Representative immunoblotting images showing the expression levels of TUJ1 (50 kD) and TH (60 kD) in iPSCs differentiated in Ctrl, CExo, and OExo for 14 or 28 days. The average optical density (OD) value of each immunoreactive band is shown above the corresponding band in the figure. β-actin (42 kD) was used as an internal control. *n* = 9 fields (3 fields per biological replicate × 3 biological replicates per group) in (**B**,**D**); *n* = 3 biological replicates per group for (**E**). Mean ± SD. ** indicates *p* < 0.01, *** indicates *p* < 0.001. Scale bar: 50 μm in (**A**,**C**).

**Figure 7 ijms-24-11048-f007:**
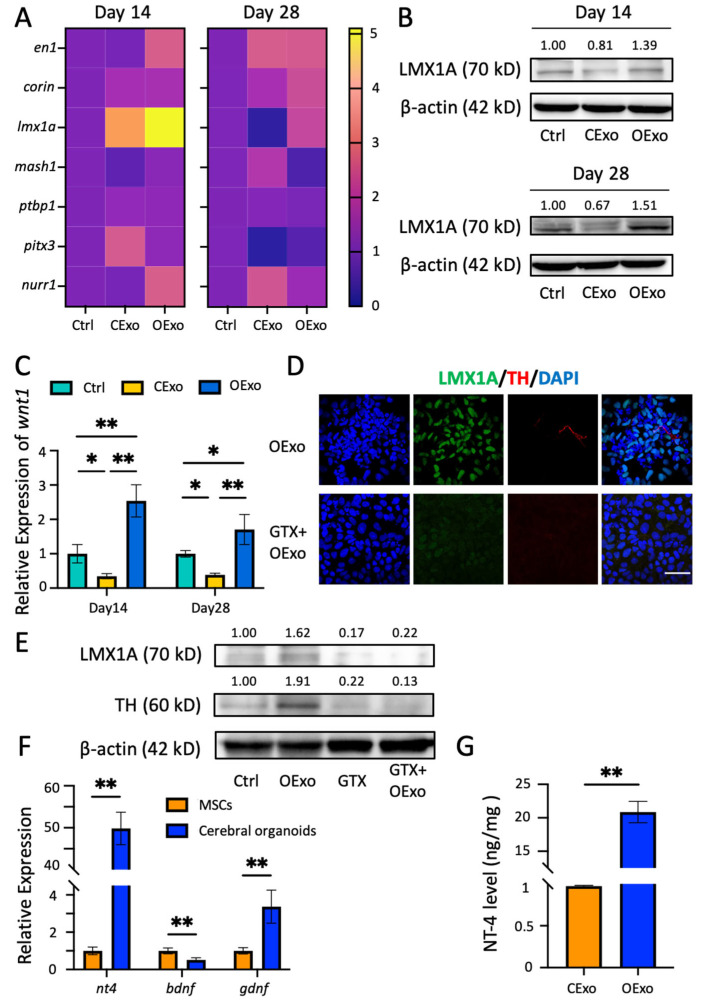
OExo promoted iPSC differentiation into DA neurons through LMX1A-dependent pathway. (**A**) Heatmaps showing the mRNA expression level of TFs, including en1, corin, lmx1a, mash1, ptbp1, pitx3, and nurr1 in iPSCs differentiated in Ctrl, CExo, and OExo for 14 or 28 days. The color scale 0–5 represents the fold change in mRNA expression level. (**B**) Representative immunoblotting images showing the expression levels of LMX1A (70 kD) in iPSCs differentiated in Ctrl, CExo, and OExo for 14 or 28 days. β-actin (42 kD) was used as an internal control. (**C**) Numerical analysis of the mRNA levels of wnt1 in iPSCs differentiated in Ctrl, CExo, and OExo for 14 or 28 days. (**D**) iPSCs differentiated in OExo or OExo together with an LMX1A-blocking peptide, GTX31507-PEP (GTX), for 14 days, followed by double immunostaining for LMX1A and TH. Note that LMX1A expression and TH production were almost completely abolished via GTX pretreatment in iPSCs differentiated with OExo. (**E**) Immunoblotting images for LMX1A (70 kD) and TH (60 kD) in iPSCs differentiated in Ctrl, OExo, GTX, and GTX with OExo for 14 days. β-actin (42 kD) was used as an internal control. Note that LMX1A expression and TH production were almost completely abolished via GTX pretreatment in iPSCs differentiated with OExo. (**F**) Numerical analysis revealed significantly enriched mRNA levels of nt-4 (~50-fold) and gdnf (~3-fold) in cerebral organoids compared with in MSCs; moreover, the bdnf mRNA level in cerebral organoids was about 1/2 of the level in MSCs. (**G**) The NT-4 content in OExo was ~20-fold that of CExo, as revealed via NT-4 ELISA. *n* = 3 independent biological replicates per group for all of the statistical analyses. Mean ± SD. * indicates *p* < 0.05, ** indicates *p* < 0.01; scale bar: 50 μm. The average optical density (OD) value of each immunoreactive band is shown above the corresponding band in the figure (**B**,**D**).

## Data Availability

The data that support the findings of this study are available from the corresponding author upon reasonable request.

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
