# Peer review of "Cerebral-Organoid-Derived Exosomes Alleviate Oxidative Stress and Promote LMX1A-Dependent Dopaminergic Differentiation"

_ijms, 2023, doi:10.3390/ijms241311048_

Round 1

Reviewer 1 Report

  • A word needs clarification; intercellular vs. intracellular (Line 49)
  • Authors are required to provide the citation of the statement in the manuscript. Refer to line 49~51.
  • What type of molecular signature authors are referring to? (Line 59)
  • Authors should provide the actual details about the exosome composition or structures rather than just stating a general statement “The composition of exosomes, particularly their surface and membrane proteins, plays a crucial role in their biological properties.
  • Authors are claiming that neural exosomes may have a positive impact on CNS diseases. Is there a scientific justification that these exosomes only impact CNS diseases and not peripheral nervous system (PNS) diseases, injuries, or trauma?

Overall the manuscript is written well. However, there are numerous minor grammatical mistakes and errors. 

Reviewer 2 Report

The paper on “Cerebral Organoid-Derived Exosomes Alleviated Oxidative 2 Stress and Promoted LMX1A-Dependent Dopaminergic Differ- 3 entiation" reports on interesting data regarding an important role of cerebral organoids (OExo) in the nerve cell viability and providing additional evidence of the effects of cerebral organoids as a neuro-protective agent on dopaminergic neurons in neurodegenerative diseases.

The study was well designed. Also, the paper was nicely written and the authors used a good range of modern technologies to corroborate their hypothesis.

Before acceptance, the authors should discuss the next issue to improve the overall quality of the manuscript: 

Comment

Basically, I accept their data and agree with their conclusion. The authors clearly demonstrated the protective potential of cerebral organoids in the nerve cell differentiation.

In addition to the oxidative stress, how about the interaction of cerebral organoids (OExo) in the inflammatory reaction in the neuron? Did the authors have any data about the direct or indirect evidence that cerebral organoids (OExo) are implicated with inflammatory degeneration of nerve tissues? If possible, from the point of view of its anti-inflammatory potential in the treatment, the authors should discuss the issue in the revised manuscript.

Reviewer 3 Report

In this manuscript, authors show the important neuroprotective role of exosomes from cerebral organoids (OExo) and mesenchymal stem cells (MSCs)-derived exosomes (CExo). In particular, they show that OExo mitigates H2O2-induced oxidative stress and apoptosis in rat midbrain astrocytes by reducing excess ROS production, anti-oxidant depletion, lipid peroxidation, mitochondrial dysfunction and apoptosis. Moreover, authors pointed out the peculiar role of OExo in promoting the differentiation of human stem cells into dopaminergic neurons, in a process involving the neurotrophic factors NT-4 and GDNF and the LMX1A transcription factor. This is an interesting original study, methodologically well performed. It is also clearly written and results support conclusions. The study provides new insight into the biological effects of cerebral organoids and highlights their potential use in the treatment of neurodegenerative diseases such as Parkinson's disease (PD).

Author Response

Reviewer 3s comments

In this manuscript, authors show the important neuroprotective role of exosomes from cerebral organoids (OExo) and mesenchymal stem cells (MSCs)-derived exosomes (CExo). In particular, they show that OExo mitigates H2O2-induced oxidative stress and apoptosis in rat midbrain astrocytes by reducing excess ROS production, anti-oxidant depletion, lipid peroxidation, mitochondrial dysfunction and apoptosis. Moreover, authors pointed out the peculiar role of OExo in promoting the differentiation of human stem cells into dopaminergic neurons, in a process involving the neurotrophic factors NT-4 and GDNF and the LMX1A transcription factor. This is an interesting original study, methodologically well performed. It is also clearly written and results support conclusions. The study provides new insight into the biological effects of cerebral organoids and highlights their potential use in the treatment of neurodegenerative diseases such as Parkinson's disease (PD).

Authors’ response: We sincerely thank the Reviewer for your time involved in reviewing the manuscript, the precise summary and positive remarks.